# Impacts of Nutrition Subsidies on Diet Diversity and Nutritional Outcomes of Primary School Students in Rural Northwestern China—Do Policy Targets and Incentives Matter?

**DOI:** 10.3390/ijerph16162891

**Published:** 2019-08-13

**Authors:** Qihui Chen, Chunchen Pei, Yunli Bai, Qiran Zhao

**Affiliations:** 1Center for Food and Health Economic Research, College of Economics and Management, China Agricultural University, Beijing 10083, China; 2Key Laboratory of Ecosystem Network Observation and Modeling, Institute of Geographic Sciences and Natural Resources Research, Chinese Academy of Sciences, Beijing 100101, China; 3UN Environment International Ecosystem Management Partnership (UNEP-IEMP), Beijing 100101, China

**Keywords:** malnutrition, nutrition subsidy, policy target, incentive, dietary structure, rural China

## Abstract

Many developing countries have implemented nutrition intervention programs to reduce child malnutrition. However, the effectiveness of these programs differs greatly, and it remains unclear what is causing the differences in effectiveness across different programs. To shed some light on this issue, this article examines the role the specificity of policy targets, along with the incentives attached, plays in affecting the effectiveness of nutrition intervention programs. More specifically, we examined how different policy targets (and the associated incentives) affect primary students’ dietary structure and (thus) their nutritional and health status by analyzing a randomized intervention in rural Northwestern China that was designed with two treatment arms. The two treatments provided the same nutrition subsidy to project students but with different policy targets, one with a specific target of “anemia reduction” and the other with a general target of “malnutrition reduction”. Our analysis revealed that compared to the treatment arm with only a general policy target, the treatment arm with the specific “anemia reduction” target was more effective at improving students’ nutritional and health status, as measured by the incidences of being anemic and underweight, presumably through helping them develop a dietary structure with more flesh meat, bean products, vegetables, and fruits.

## 1. Introduction

Child malnutrition is commonly observed among school-age children in many low-income countries and regions [1,2,3]. Not only can child malnutrition lead to severe health problems among school-age children, but it may also impose serious constraints on their development of cognitive skills [4,5,6]. For example, anemia caused by iron deficiency can cause children to feel tired and weak, thereby preventing them from concentrating on their coursework [7,8]. These problems, in turn, undermine these countries and regions’ human capital formation for future development [9,10,11].

To fight against child malnutrition, many developing countries have implemented nutrition intervention programs targeting poor children at school. However, empirical evidence accumulated thus far has largely failed to depict a conclusive picture of the effectiveness of these interventions. Consider school snack programs, for example. While a study conducted in Bangladesh found that providing fortified snacks at schools increased children’s BMI by 4.3% [12], another study done in Colombia found that a similar snack program led to no significant difference in (the change of) children’s BMI between treatment and control schools [13]. Similar discrepancies exist among school meal programs. In Jamaica, Powell et al. [14] found that children who received in-school meals gained a significant amount of weight and became taller than those in the control group, whereas Singh et al. [15] found in India that a similar intervention has no significant impact overall—only among a very particular subset of children, i.e., those whose families had recently suffered from droughts, were some positive effects (on weight and height) found. These discrepancies in existing findings render it difficult to generalize lessons from these interventions, thus imposing great challenges on effective policy making. Clearly, to better inform policy, a crucial first step is to gain a deeper understanding of the underlying causes of these discrepancies.

While providing an ultimate understanding of the causes of these discrepancies is very difficult (if not entirely impossible), the present study intends to shed some light on this issue by analyzing a school-based randomized intervention implemented in rural Northwestern China, which aimed to reduce child malnutrition through different treatments (more details are provided in Section 2 below). Our analysis seeks to answer two questions. First, how does the provision of nutrition subsidies translate into observed nutritional and health outcomes? Most of the existing nutrition intervention programs did not closely monitor whether and how the nutrition supplements (or subsidies) provided to the targeted recipients were consumed (or spent) by them. Instead, these studies tend to rely on macro-level nutritional outcomes (e.g., height- and weight-for-age z-scores), usually measured at the end of the program, to assess the impacts of these interventions. However, even given the same type of intervention (or same amount of subsidy), the targeted students (and/or school administrative staff) may react differently when provided with different policy targets and incentives. To the extent that these macro-level nutritional outcomes were largely determined by the actual amount and the structure of food consumed by the targeted students, different behavioral/dietary responses of these students (and/or of school administrative staff) may lead to vastly different observed outcomes. To provide an answer (at least a partial one) to this question, we exploit information on sampled students’ 24-h food-intake recall (which involves more than 30 food categories) recorded during the project to infer how foods provided by the intervention were consumed by those treated children.

The second question we seek to answer is: Will policy targets with different levels of specificity, for example, general ones such as “malnutrition prevention” and specific ones such as “anemia reduction”, lead to different behavioral responses and (thus) nutritional and health outcomes, partly through different incentives attached to these policy targets (since certain incentives are presumably needed to achieve any specific policy target)? For example, stipulating a specific policy target (say, anemia reduction) may render the treatment more operable and its impacts more trackable, but pursuing such a target may dilute resources that can be used to achieve other unspecified yet desirable goals. The combined effect is theoretically ambiguous and is thus ultimately an empirical question. The two-treatment design of our intervention program helps provide an answer to this question. Exploiting that design, we compare the impacts on students’ dietary structure and their nutritional and health outcomes of a treatment arm that involves a nutrition subsidy with only a general target of “malnutrition reduction” and the impacts of another treatment with the same subsidy but specifying a policy target directly tied to “anemia reduction”. Our analysis finds that the treatment arm with a specific target of “anemia reduction” (as well as the incentives attached to it) was more effective at reducing malnutrition, presumably through enhancing students’ diet diversity. These findings offer experiences for other low-income countries and regions to reduce malnutrition among children.

The remainder of the article proceeds as follows. The next section describes the intervention and data as well as the method underlying the estimation of the impacts of nutrition subsidies (with different policy targets and incentives) on students’ dietary intake and nutritional status. Section 3 presents our findings and discusses their implications and limitations. Section 4 concludes the study.

## 2. Data and Methods

### 2.1. Study Area

The intervention program analyzed in this study was implemented in 59 primary schools selected from rural areas of two provincial-level administration units in China, namely, Qinghai Province and Ningxia Autonomous Region, during the period of October 2009–May 2010. Endowed with disadvantaged geographical locations and natural conditions, these two provinces are relatively underdeveloped compared to other provinces in China. Child malnutrition is also prevalent. For example, a study conducted around the time of our intervention found that in 2009, 25.4% of primary-school students were anemic in rural areas of Ningxia [16]. Furthermore, the baseline survey of our project discovered that among the 6994 primary school students involved in the project, who came from 10 rural counties in Qinghai and Ningxia, 19.5% were anemic and 7.7% were underweight according to WHO standards [17,18] (the cut-off values of Hb concentration in blood in grams per liter (g/L) provided to determine whether an individual is anemic is 115 g/L for children aged 5–11 years and 120 g/L for children aged 12–14, irrespective of their gender; children whose BMI is below −2 standard deviations (SDs) of the BMI-for-age z-score distribution are considered as being underweight [17,18]).

### 2.2. Sampling

The intervention targeted 4th and 5th graders enrolled in elementary schools in rural Qinghai and Ningxia. The study received ethical approval from the Stanford University Institutional Review Board (IRB) on 21 July 2009 (protocol number: 17071). All necessary permissions were also obtained from the Chinese government. All participating students, as well as their parents, gave their written assent to participate in the project; their legal guardians in school (the schoolmasters and the headteachers) also provided written consents. All project participants were aware of the (minimal) risks involved and understood that their participation was entirely voluntary.

Based on the distribution of per capita gross values of industrial output [19], about 30 townships from each province were randomly selected. In each township, only those schools offering 4th- and 5th-grade classes and accommodating at least 400 students were selected. In total, 59 schools and 6994 students from 4th and 5th grades of these schools were selected to participate in the project. Information on dietary patterns and personal and family characteristics was collected from all students, yet due to budgetary constraints, information on hemoglobin (Hb) concentration in blood, as well as other physical health indicators (e.g., height and weight), was collected from only about half of the sample.

To fully capture the impacts of the treatments, our analysis focuses only on the 2199 boarding students in the sample (of 6994 students). Note that since boarding students are not free-living, their meal plans are almost entirely controlled by their schools, in which case their food-consumption activities reflect closely their school food-purchase activities. In contrast, non-boarding students can always have their meals (especially dinners) at home and thus presumably received much less influence of the project than their boarding counterparts. As mentioned above, due to budgetary reasons, while dietary information was collected from all (boarding) students, health information was collected only from about half of them: 1020 boarding students have information on Hb concentration level, 952 have height and weight information, and 866 have both sets of information. Thus, the final analytical sample is comprised of 866 boarding students with information available on all three dimensions: (1) dietary patterns, (2) anemia-related outcomes (i.e., Hb concentration and incidence of anemia), and (3) overall nutritional status (e.g., height, weight; thus BMI and the incidence of being underweight).

### 2.3. Intervention

At the beginning of the intervention in October 2009, the 59 project schools were randomly assigned into three groups: 29 schools in the control group, 15 schools in treatment group 1, and 15 schools in treatment group 2. Students in the targeted grades (i.e., 4th and 5th grades) in the treatment groups received continuous interventions from November 2009 through May 2010, with a one-month pause during the winter break in February 2010. The control group received no intervention during the entire intervention period.

Treatment group 1 received a nutrition subsidy with a general policy target of “malnutrition reduction”. The monetary equivalent of the total amount of nutrition subsidy provided to each school in this group was 225 Yuan (=33 U.S. dollars) per enrolled student (which cost 1.5 yuan/day, enough for purchasing 60 g of red meat, for a period of 150 days). The subsidy money was transferred to the bank account of each school and the schoolmasters were able to use this money for nutrition-related expenses (in any way they deemed reasonable). In addition, each schoolmaster in this treatment group was informed about the main aim of the intervention project (i.e., to reduce child malnutrition) and was given three pieces of additional information: (a) the share of enrolled students who were anemic (not the specific individuals but the average rate of the whole school), (b) descriptions of effective methods for reducing iron-deficient anemia, and (c) details about anemia’s relation with school attendance, educational performance, and cognitive development. However, the project team did not provide any specific instructions or stipulate specific requirements on what foods the schools should purchase; schoolmasters were allowed to make their own decisions on how the subsidy money may be spent to achieve the goal of malnutrition reduction.

Schools in treatment group 2 received a nutrition subsidy identical to the one provided to treatment group 1, plus a specific policy target of “anemia reduction”. To help achieve this target, an incentive in the form of a potential monetary bonus was provided to schoolmasters in this group; the amount of the bonus was tied to actual reductions in anemia prevalence among students in their schools—more specifically, a schoolmaster would receive a 150-*Yuan* (or 22-U.S. dollar) bonus for each of his or her students whose status changed from being anemic to being non-anemic over the course of the intervention. As with schoolmasters in treatment group 1, those in treatment group 2 were informed about the main aim of the intervention project (i.e., to reduce child malnutrition) and were provided with the same three pieces of anemia-related information. Similarly, they were allowed to make their own decisions on how the subsidy money may be spent; no requirements were imposed by the project team or the local government.

Two features of the intervention are worth noting. First, since the bonus provided to schoolmasters in treatment group 2 would not be realized until the end of the intervention period when the actual reductions in anemia prevalence have been revealed (note that it may not even be realized if there are no reductions in anemia prevalence), the actual amount of subsidy per student received by the two treatment groups (i.e., the amount that can be used for food purchase in per student terms) was identical during the intervention. Second, treatment 1, although with no specific policy target specified, is not without incentives attached—schools in this group always had the incentive to perform well because the project was supported financially and logistically by the local governments. The differences in the level of specificity of policy targets and the incentives attached to them are likely to generate differences in motivation and food purchase behavior among schoolmasters in the two treatment groups, which may in turn lead to different impacts on students’ dietary structure and nutritional outcomes.

To help assess whether the randomized group assignments were done properly, Appendix A
Table A1 presents the school-level means of a set of baseline characteristics (for both the full sample and the boarding sample), including indicators of students’ dietary structure, and their nutrition and health outcomes (Panel A), personal and family characteristics (Panel B), and school characteristics (Panel C), separately for the three groups. As can be seen in the Table, most of these variables are quite balanced across groups, with minor differences due to sampling errors, suggesting that the random group assignments were done reasonably well.

### 2.4. Outcomes of Interest

The outcomes of interest of this study include students’ (i) anemia-related outcomes (i.e., Hb concentration level, measured by medical professionals, and the incidence of anemia, computed according to WHO standards [17,18]), (ii) their overall nutritional status outcomes (i.e., BMI z-scores and the incidence of underweight, converted using height and weight information provided by medical professionals), and (iii) dietary structure (measured by the consumption of specific food categories, recorded in a 24-h dietary recall, and an overall dietary diversity index constructed from this food-consumption information, detailed in Section 2.5); data regarding all of which were collected through the baseline and the follow-up surveys. The first set of outcomes (i.e., anemia-related outcomes) are natural for assessing the effectiveness of the intervention, by capturing its impact on the intended outcomes—if the specific policy target of “anemia reduction” (and the incentives attached) did work, one would expect to see at least some reductions in the prevalence of anemia among students in treatment group 2. The second set of outcomes (i.e., overall nutritional status) help to assess whether these changes (in intended outcomes) translated into improvements in their overall nutritional status by capturing the impacts of the intervention on potential unintended outcomes (spillovers). The third set (dietary structure) helps us capture the possible working channels, as it recorded students’ food-consumption activities (and reflected schools’ food-purchase activities) during the intervention (Figure 1).

The surveys also collected information on sample students’ personal characteristics (e.g., age, sex and sibship size), family characteristics (e.g., parental education, parental migration status, and household assets), and school characteristics (e.g., student-teacher ratio and teacher qualifications), which are used as covariates in our statistical analysis. Summary statistics of all these variables are presented in Appendix A
Table A1.

### 2.5. Measuring Students’ Dietary Structure

Of particular importance for the purpose of this study, information on a 24-h dietary recall on students’ consumption of 33 specific food items was recorded in both the baseline survey (October 2009) and the follow-up evaluation survey (May 2010), which helps provide some information on how the nutrition subsidy was spent. During the surveys, students recorded their recall on consumption frequencies of these 33 food items during the previous 24 h, with possible help from their teachers and trained enumerators. Although their recall was not guided or helped by professional nutritionists, the fact that these students were boarding (i.e., not free-living) students helps strengthen the quality of their call data, in that the project team were aware of their possible dietary choices.

The wealth of information recorded in the dietary recall is then exploited to construct a dietary diversity score (DDS) that has been used in many recent studies to measure school-age children’s dietary structure in the context of China [20,21,22,23]. The DDS is constructed based on a set of guidelines provided by the FAO, i.e., “Guidelines for Measuring Household and Individual Dietary Diversity (Version 4)” [24]. More specifically, the DDS counts the number of food categories, e.g., “grains” and “fish”, that an individual consumed during the past 24 h, out of a set of FAO-recommended food categories. A DDS closer to the number of the FAO-recommended food categories indicates a more diverse diet structure (that is consistent with FAO recommendations).

The original FAO guidelines [25] involve 14 food categories (Table 1, column 1), but given data availability and local conditions we modified the construction algorithm of the DDS slightly: first, we combined 6 FAO-specified food categories into 3 larger ones—“vitamin A rich vegetables and tubers” and “white tubers” into “tubers”, “dark green leafy vegetables” and “other vegetables” into “vegetables”, and “vitamin A rich fruits” and “other fruits” into “fruits”; second, we replaced two categories with their local counterparts (e.g., we replaced “legumes” with “bean products”); and finally, we dropped one category (“oil and fat”) from the FAO list (Table 1, column 2).

### 2.6. Estimation Method

As previously noted, our study intends to analyze how treatments with different policy targets (and with different embedded incentives) may lead to different nutritional and health outcomes, through their impacts on students’ dietary pattern/structure (Figure 1).

Since random group assignments help eliminate selection bias due to unobserved factors [26], the impacts of the nutrition interventions described above on the outcomes of interest can be estimated by applying ordinary least-squares (OLS) regressions to data collected in the follow-up survey (the evaluation period):
*Oeval_ji_* = β_0_ + β_1_*T1_i_* + β_2_*T2_i_* + ε*_ji_*,(1)
where the outcome variable, *Oeval_ji_*, is the *j*th dietary, nutritional or health outcome (e.g., the DDS or anemia-related outcomes) of student *i*, observed in the evaluation period (May 2010); *T1_i_* and *T2_i_* indicate, respectively, student *i*’s treatment status, with *T1_i_* (*T2_i_*) = 1 indicating that student *i* received treatment 1(2) and *T1_i_* (*T2_i_*) = 0 otherwise. If the randomization was done properly, the parameters β_1_ and β_2_ capture the impacts of treatment 1 (i.e., “nutrition subsidy + general target (malnutrition reduction)”) and treatment 2 (i.e., “nutrition subsidy + specific target (anemia reduction)”) on the *j*th outcome being examined.

However, due to the modest number (i.e., 59) of project schools, some student or school characteristics may not be perfectly balanced across the three groups (Appendix A
Table A1), even under randomized group assignments. To address this issue, we modify the estimating equation (1) in two ways. First, we adopt a difference-in-differences setup, replacing the dependent variable *Oeval_ji_* in Equation (1) with ∆*O_ji_* = *Oeval_ji_* − *Obase_ji_*, i.e., the change in the *j*th outcome of interest from its baseline value, *Obase_ji_* (observed in October 2009) to its evaluation-period value, *Oeval_ji_* (observed in May 2010). This modification eliminates the influences of all fixed factors, observed or unobserved, as well as the potential intra-class correlation caused by these fixed factors. Second, we include a set of baseline characteristics (X) observed at the student level (e.g., gender, age, and ethnicity), the household level (e.g., parental education, migrant status, and asset holding), and the school level (e.g., student-teacher ratio and the proportion of senior-teaching staff), in the model to help control for undesired pre-treatment differences between the treatment and control groups:
Δ*O_ji_* = β_0_ + β_1_*T1_i_* + β_2_*T2_i_* + X*_i_*δ + ε*_ji_*(2)

Unless otherwise stated, all estimates presented below are obtained after controlling for the full set of covariates reported in Appendix A
Table A1. All statistical analyses were performed using STATA 14, and results with *p*-values < 0.05 are considered statistically significant.

## 3. Results and Discussions

### 3.1. Descriptive Patterns of Outcome Variables

Table 2 presents summary statistics of outcome variables measured in both the baseline (October 2009) and the follow-up evaluation (May 2010) periods, separated for the three groups involved in the project. Simple comparisons across the columns of the Table suggest improvements in students’ nutritional and health outcomes: compared to students assigned to the control group, those assigned to the treatment groups in general performed better in both overall nutritional status (weight-related outcomes) and anemia-related outcomes in the evaluation period, although not necessarily so at the baseline. Increases in dietary diversity may play a role in driving these improvements: compared to their counterparts in the control group, students in the treatment groups consumed more frequently about half of the foods involved in the DDS (e.g., flesh meat, vegetables, fruits, bean products, and dairy products) in the evaluation period. Note that because these descriptive results are obtained without netting out the influences of potential confounding factors, they can only be considered as suggestive. More credible results, which are obtained by controlling for a large set of covariates (reported in Appendix A
Table A1), are presented and discussed in the following subsections.

### 3.2. Impacts on Nutritional Outcomes

Focusing on the impacts of the nutrition intervention on (boarding) students’ nutritional outcomes, Table 3 reports the impacts on two set of outcomes: the first includes two anemia-related outcomes (Hb concentration level and the incidence of anemia); the second includes two measures of overall nutritional status (BMI z-scores and the incidence of being underweight). As expected, treatment 2 (with the specific target of anemia reduction) was indeed more effective at improving students’ anemia-related outcomes (Table 3, columns 1 and 2). More specifically, while the treatment of “nutrition subsidy + specific target (anemia-reduction)” (treatment 2) raised students’ Hb concentration by 4.49 g/L (column 1), which translated into a 12 percentage-point reduction in the prevalence of anemia among them (column 2), the treatment of “nutrition subsidy + general target (malnutrition-reduction)” (treatment 1) had essentially no impact. The difference between the impacts of the two treatments is statistically significant at the 5% level (see the last row of Table 3 for the associated *p*-values).

A similar pattern is observed for overall nutritional status (weight-related outcomes). While the treatment of “nutrition subsidy + specific target (anemia-reduction)” (treatment 2) raised treated students’ BMI z-scores by 0.12 SDs (column 3), which in effect lowered their probability of being underweight by 4.1 percentage points (column 4), the impacts of the treatment of “nutrition subsidy + general target (malnutrition-reduction)” (treatment 1) are smaller, and are not statistically significant at any conventional level (the difference between the impacts of the two treatments is statistically insignificant, though).

As shown in Appendix A
Table A2, none of the personal or household characteristics has statistically significant independent power to help explain the baseline-endline changes in these nutritional and health outcomes.

### 3.3. Impacts of Nutrition Interventions on Dietary Structure

To gain an understanding of why the impacts of the two treatments differed, we further estimate the impacts of the two treatment arms on students’ dietary structure (which recorded students’ food-consumption activities during the intervention), measured by the DDS detailed in Section 2.5. As shown in column (1) of Table 4, while both treatment arms are quite effective at improving students’ diet diversity (measured by the DDS), the impact of the treatment of “nutrition subsidy + specific target (anemia reduction)” (treatment 2) is about 1/3 larger than that of the treatment of “nutrition subsidy + general target (malnutrition reduction)” (treatment 1). More specifically, while the latter raised the total number of food categories an average student consumed a day by 0.96 (out of the 10 FAO-recommended food categories involved in the DDS), the former raised this number by 1.26 (out of 10 FAO-recommended categories).

Given the baseline DDS of slightly above 5.0 (out of 10 FAO-recommended categories) (Appendix A
Table A1), these impacts amount to non-trivial 1/5 to 1/4 increases in diet diversity. To further examine which food categories were most affected by the treatments, we estimated their impacts separately on the consumption incidence of each of the 10 food categories involved in the DDS. The main results are reported in columns (2)–(11) of Table 4. For ease of comparison, the columns of this table have been organized, from left to right, in descending order of the impacts of the “nutrition subsidy + general target (malnutrition reduction)” treatment (treatment 1). Two results are notable. First, the intervention (in particular, the “nutrition subsidy + specific target (anemia reduction)” treatment) significantly raised students’ consumption of 6 out of the 10 food categories involved in the DDS. Second, students in the “nutrition subsidy + specific target (anemia reduction) treatment group consumed more than their counterparts in the “nutrition subsidy + general target (malnutrition reduction)” group in 5 out of 6 of these food categories, the exception being the “other meat” group (column 2).

Comparisons of the impacts of the two treatment arms on student’s dietary pattern, reported in Table 4, suggest that the anemia-remedying impact is likely driven by the increased consumption of flesh meat, bean products, vegetables, and fruits by students assigned to the “nutrition subsidy + specific target (anemia reduction)” treatment group. As nutritionists have long discovered, red meat, bean products (e.g., tofu), and leafy greens are good sources of (nonheme) iron, the key element to avoid anemia [26]. Some leafy greens such as collard greens also contain folate, the intake of which may help prevent folate-deficiency anemia [27,28]. The rich vitamin C contained in citrus fruits and vegetables also helps enhance iron absorption [29]. Interestingly, not only did students assigned to the “nutrition subsidy + specific target (anemia reduction)” group (treatment group 2) consume more of the aforementioned anemia-remedying foods, but they were also more likely to consume more foods that are not directly anemia-remedying such as dairy products. This suggests that the specific target of “anemia reduction” (and the incentives attached) may serve to generate some unintended yet desirable outcomes (i.e., positive spillovers).

### 3.4. Caveats

A number of caveats should be kept in mind when interpreting the above findings. First, since we only have data on whether a student consumed food in a given DDS category in the previous 24 h, rather than the actual amount of food consumed in a longer time period, our findings regarding students’ dietary patterns may only be considered as “indirect” evidence of how the two treatments worked. More direct and accurate evidence may be found in datasets collected using more detailed questionnaires and/or diaries. Collaboration with nutritionists may also enable the measurement of students’ energy intakes (e.g., calories from protein, fat, carbohydrate, etc.), so that a more comprehensive picture of their consumption pattern can be depicted. Another possible direction of future research (as suggested by one of the reviewers of this article) is to divide food groups by key nutrients involved, such as iron-rich foods and vitamin A-rich foods, when examining students’ dietary structure from the perspective of nutrition intake.

Second, our intervention involved only two treatments. Since the two treatments differ in both (the level of specificity of) policy targets and the incentives attached, it is not possible to isolate the impact of one from the other using available data; only the combined effects of policy targets and incentives are identifiable. We are also unable to examine whether specifying other policy targets, such as those aiming to increase vitamin A or protein consumption, can yield impacts that are even more effective. Thus, more research may be needed to examine the role different specific policy targets play in driving the effectiveness of nutrition interventions, as well as the underlying mechanisms (e.g., schoolmasters’ behavioral responses). In any case, however, the randomized design of our intervention project helped provide credible evidence that at least in the context of rural Northwestern China, where anemia is prevalent, nutritional inventions with specific targets of anemia reduction (with proper incentives attached to them) can be quite effective at reducing the prevalence of anemia and that of malnutrition simultaneously.

Third, some special features in our study area may have helped drive our main findings. For example, the study area is characterized by a high prevalence of anemia among school-age children (nearly 20%) [16]. Compared to other low-income regions with relatively lower anemia rates, our treatment 2 may have a larger impact on anemia reduction due to the “law of diminishing returns” (in economics). The high proportion of boarding students in our project schools (>30%) may also help strength the impacts of nutrition subsidies, because the boarding arrangement ensures that almost all the food consumed by boarding students came from the intervention. Finally, extreme poverty in the study area may also play a role in that the same monetary bonus would be deemed more valuable in poorer regions, which would provide stronger incentives to schoolmasters of the project schools. Thus, to ultimately pin down the roles of policy targets and incentives, one has to consider the influences of these area- and context-specific factors. Cross-regional studies that control for region-specific features (e.g., meta-analyses) are likely to be fruitful in this regard.

## 4. Conclusions

By comparing the impacts of two treatments involved in a randomized intervention conducted in rural Northwestern China, this paper provides a potential explanation of why the effectiveness of previous nutrition intervention programs varies: specifying a concrete policy target (and attaching appropriate incentives to it) may help increase the effectiveness of the program, even if this target reflects only one dimension of the overall goal of malnutrition reduction. Our statistical analysis finds that the treatment with a specific target of “anemia reduction” was more effective at reducing malnutrition—not only did it significantly reduce the prevalence of anemia among targeted students (by about 12 percentage points), but it also reduced the prevalence of underweight among these children (by about 4 percentage points), thereby generating (unintended) positive spillovers. In contrast, the treatment with only a general target of “malnutrition reduction” had little impact on either outcome. Further analysis suggests that the impacts of the treatment with a specific target worked through improving students’ dietary structure—it was more effective at helping the targeted students develop a more diverse diet, a diet with not only more foods that are good sources of iron (e.g., flesh meat, leafy vegetables, and bean products) but also more foods that are not major sources of iron (e.g., dairy products).

There are a number of reasons for why the treatment with a specific policy target (i.e., anemia reduction) performed better. Stipulating a specific policy target may render the treatment more operable and its impacts more trackable. When this specific policy target is tied to certain incentives, these incentives become more transparent and thus more credible, thereby helping to enhance the effectiveness of the intervention. The actions taken to achieve the specified policy target may also generate some positive spillovers. As discussed above, the “anemia reduction” target provided students in the “nutrition subsidy + specific target (anemia reduction)” group with more foods that are not even anemia-remedying.

In closing, a caution against possible unintended effects is in order. While our findings suggest that the treatment with the “anemia reduction” target performed better than the one with the general “malnutrition reduction” target, the specific target and the associated monetary bonus may create unintended and undesirable incentives. For example, unscrupulous schoolmasters might provide extra iron-rich foods to only the anemic students, but not other students in their schools; some schools might even offer only iron-rich foods, but not other nutritious foods (e.g., dairy products), to these students, which in effect undermines their diet diversity and their nutritional and health status. To help prevent these undesirable behaviors in our project, we withheld information on the names of individual students who were anemic from the schoolmasters, only informing them the average anemia rate within their schools. However, it may be difficult to foresee (and thus difficult to prevent beforehand) unintended effects when the policy target being considered is more complex, such as “underweight reduction”, which may be achieved undesirably, e.g., by providing plenty of high-calorie foods to malnourished children, if improper incentives were provided. In any case, alternative policy targets, proper incentives, and potential working channels must all be fully considered before specific interventions are implemented.

## Figures and Tables

**Figure 1 ijerph-16-02891-f001:**
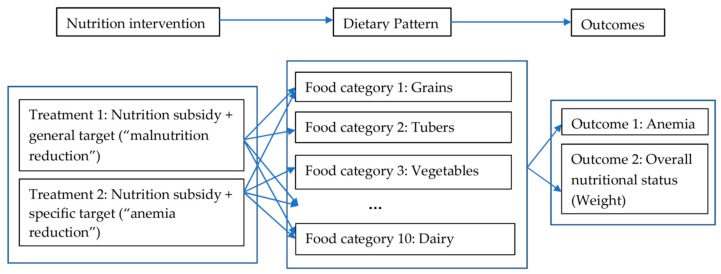
Conceptual framework.

**Table 1 ijerph-16-02891-t001:** Food categories used in the construction of dietary diversity scores (DDS).

(1) Food Categories Used to Construct the DDS	(2) Food Categories Involved in FAO Guidelines
Grains	Grains
Tubers	Vitamin A rich vegetables and tubers; White tubers
Vegetables	Dark green leafy vegetables; Other vegetables
Fruits	Vitamin A rich fruits; Other fruits
Flesh meat	Flesh meat
Other meat	Organ meat
Eggs	Eggs
Fish	Fish
Bean products, nuts and seeds	Legumes, nuts and seeds
Milk and milk products	Milk and milk products
n/a	Oil and fat

Notes: 1. The FAO guidelines (column 2) were provided by the FAO [25]. Given data availability and local conditions, we combined “dark green leafy vegetables” and “other vegetables” into “vegetables”, “vitamin A rich vegetables and tubers” and “white tubers” into “tubers”, and “vitamin A rich fruits” and “other fruits” into “fruits”. We also replaced “organ meat” with “other meat” and “legumes” with “bean products” and dropped “oil and fat” (column 1).

**Table 2 ijerph-16-02891-t002:** Summary statistics of variables measured before and after the intervention.

Outcomes	(1) Control Group Mean (SD)		(2) Treatment Group 1 Mean (SD)		(3) Treatment Group 2 Mean (SD)	
	Baseline Values	Endline Values	Baseline Values	Endline Values	Baseline Values	Endline Values
Anemia-related outcomes:						
Hb concentration (g/L)	128.03 (12.95)	127.93 (14.86)	128.51 (12.63)	128.11 (15.86)	127.84 (12.80)	130.95 (15.66)
Anemia (=1 if yes)	0.22 (0.42)	0.23 (0.42)	0.18 (0.38)	0.22 (0.42)	0.23 (0.42)	0.16 (0.36)
Overall nutritional status:						
BMI-for-age z-scores	−0.68 (0.94)	−0.76 (0.97)	−0.70 (0.91)	−0.71 (0.95)	−0.63 (0.91)	−0.60 (0.89)
Underweight (=1 if yes)	0.08 (0.26)	0.11 (0.32)	0.07 (0.25)	0.07 (0.26)	0.06 (0.24)	0.06 (0.23)
*Dietary structure:*						
Diet Diversity Scores	5.33 (2.32)	4.82 (2.36)	4.75 (2.17)	5.21 (2.18)	4.65 (2.20)	5.32 (2.09)
Consumption of food categories (=1 if yes):						
Grains	0.95 (0.21)	0.97 (0.18)	0.95 (0.21)	0.98 (0.13)	0.93 (0.26)	0.97 (0.17)
Tubers	0.71 (0.45)	0.75 (0.44)	0.72 (0.45)	0.74 (0.44)	0.65 (0.48)	0.65 (0.48)
Vegetables	0.83 (0.37)	0.81 (0.40)	0.71 (0.45)	0.79 (0.40)	0.71 (0.46)	0.88 (0.32)
Fruits	0.75 (0.44)	0.47 (0.50)	0.67 (0.47)	0.56 (0.50)	0.69 (0.46)	0.59 (0.49)
Flesh meat	0.54 (0.50)	0.46 (0.50)	0.49 (0.50)	0.63 (0.47)	0.46 (0.50)	0.65 (0.48)
Other meat	0.27 (0.44)	0.22 (0.42)	0.23 (0.42)	0.34 (0.47)	0.20 (0.40)	0.28 (0.45)
Eggs	0.31 (0.46)	0.34 (0.48)	0.24 (0.43)	0.33 (0.27)	0.24 (0.43)	0.30 (0.46)
Fish	0.11 (0.32)	0.13 (0.33)	0.10 (0.30)	0.08 (0.27)	0.09 (0.28)	0.09 (0.29)
Bean products, nuts and seeds	0.47 (0.50)	0.38 (0.49)	0.41 (0.49)	0.44 (0.50)	0.42 (0.50)	0.51 (0.50)
Milk and milk products	0.38 (0.49)	0.29 (0.45)	0.23 (0.42)	0.32 (0.47)	0.25 (0.44)	0.40 (0.49)
Number of observations (boarding students only)	437		219		210	

Source: Author’s survey. Notes: 1. Boarding students only. 2. Standard deviations in parentheses.

**Table 3 ijerph-16-02891-t003:** Impacts of nutrition treatments on students’ nutritional outcomes and dietary structure.

Variables	(1)	(2)	(3)	(4)
	Anemia-Related Outcomes		Overall Nutritional Status	
Changes in:	Hemoglobin Concentration (g/L)	Anemia (=1 if Yes)	BMI-for-Age z-Scores	Underweight (=1 if Yes)
Treatment 1: Nutrition subsidy	0.512	−0.005	0.080	−0.032
+ general policy target (malnutrition reduction)	(1.348)	(0.048)	(0.058)	(0.024)
Treatment 2: Nutrition subsidy	4.490 ***	−0.120 ***	0.123 ***	−0.041 *
+ specific policy target (anemia reduction)	(1.241)	(0.046)	(0.047)	(0.022)
Control variables	yes	yes	yes	yes
Observations	866	866	866	866
R^2^	0.083	0.048	0.055	0.018
*p*-values for testing the difference between the impacts of Treatment 1 and Treatment 2	0.009	0.035	0.456	0.751

Notes: 1. Control variables include ethnicity dummies (for Han, Hui, Salar, Tibetan, Tu, and “others” that are very small in numbers), a gender dummy, a grade dummy, number of siblings, father’s education (years), mother’s education (years), whether one’s mother is a migrant worker, whether one’s mother is a migrant worker, student-teacher ratio, the proportion of senior-level teaching staff, and province dummies. 2. Results with the full set of covariates are reported in Appendix A
Table A2. 3. Robust standard errors in parentheses. 4. *** *p* < 0.01, * *p* < 0.1.

**Table 4 ijerph-16-02891-t004:** Estimated impacts of treatments on students’ dietary structure.

Outcome Variables	(1)	(2)	(3)	(4)	(5)	(6)	(7)	(8)	(9)	(10)	(11)
Changes in:	DDS	Milk and Dairy Products	Other Meat	Flesh Meat	Fruits	Bean Products, Nuts and Seeds	Vegetables	Eggs	Grains	Tubers	Fish
Treatment 1: Nutrition subsidy + general target	0.956 ***	0.185 ***	0.179 ***	0.172 ***	0.166 ***	0.107 *	0.063	0.069	0.022	0.011	−0.018
(malnutrition reduction)	(0.255)	(0.057)	(0.054)	(0.053)	(0.059)	(0.060)	(0.047)	(0.054)	(0.019)	(0.051)	(0.033)
Treatment 2: Nutrition subsidy + specific target	1.263 ***	0.236 ***	0.146 ***	0.257 ***	0.196 ***	0.179 ***	0.189 ***	0.013	0.033	−0.004	0.018
(anemia reduction)	(0.224)	(0.053)	(0.048)	(0.056)	0.166 ***	(0.057)	(0.045)	(0.051)	(0.025)	(0.054)	(0.033)
Control variables	yes	yes	yes	yes	yes	yes	yes	yes	yes	yes	yes
Observations	866	866	866	866	866	866	866	866	866	866	866
R^2^	0.078	0.055	0.034	0.067	0.035	0.048	0.086	0.020	0.011	0.020	0.031

Notes: 1. Control variables include ethnicity dummies (for Han, Hui, Salar, Tibetan, Tu, and “others” that are very small in numbers), a gender dummy, a grade dummy, number of siblings, father’s education (years), mother’s education (years), whether one’s mother is a migrant worker, whether one’s mother is a migrant worker, student-teacher ratio, and proportion of senior-level teaching staffs and province dummies. 2. Robust standard errors in parentheses. 3. *** *p* < 0.01, ** *p* < 0.05, * *p* < 0.1.

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
