# Peer review of "Impacts of Nutrition Subsidies on Diet Diversity and Nutritional Outcomes of Primary School Students in Rural Northwestern China—Do Policy Targets and Incentives Matter?"

_ijerph, 2019, doi:10.3390/ijerph16162891_

Round 1

Reviewer 1 Report

Excellent research. Some corrections in English grammar are required.

Author Response

Thanks for the helpful instructions for making the writing more formal. We have revised the manuscript following your suggestions.

Reviewer 2 Report

Manuscript: ijerph-527417

Nutrition Subsidy, Diet Diversity and Nutritional  Outcomes of Primary School Students in Rural  Northwestern China – Do Policy Targets and 4 Incentives Matter?

Qihui Chen, Chunchen Pei, Yunli Bai and Qiran Zhao

General comments:

The study describes content, evaluation and results of an intervention study conducted in two districts of mainland of China between 2009/2010 in 866 school-age children. It seems that aim of the study was to evaluate results of an intervention targeting malnourished children to improve nutrition status, dietary diversity and Hb status. It is challenging to understand the sampling procedure of included children, the intervention content and implementation, and measurements. Also the statistical analysis is not appropriate. The conclusions the authors draw are not supported by the results. Also, the manuscript is missing a clear structure; please read and adhere to the instruction for authors carefully.

Major comments:

The Introduction is lacking the main aim of the study and the underlying reasons why the authors conducted this intervention, e.g. prevalence of childhood malnutrition or anemia. Instead the authors summarize already (misplaced!) conclusions in this section. On the contrary, the relevant information in the Methods section is vastly limited, and also repeated a lot of text from the Discussion and Conclusion section. Besides, the results only compared the differences between two treatment groups and control group, no statistical comparison were conducted within  two treatment groups, which limited the interpretation of the results. Moreover, the Discussion should discuss the findings and implications in the broadest context, as well as the limitations and future research direction, this part is really weak in the manuscript. The Conclusions section for IJERPH manuscripts is mandatory, so the headings of your last two sections should be “Results and Discussions” and “Conclusions”.

Specific comments:

Title

Nutrition subsidy is an intervention strategy rather than outcome in the study.

Line 17

After I went through the whole paper, I feel “The role specifying specific policy” is not accurate to describe the intervention strategies you mentioned below. The word of “Policy” is too broad, “incentives” seems to be more suitable.

Line 19

About the sentence “Two different treatment arms, one with a specific policy target (i.e. “anemia reduction”) and the other without”, it sounds like the other treatment arms didn’t undertake any intervention, how could it be called “treatment arms”? Please be more specific what exactly the intervention was.

Line 22

The results didn’t report the comparisons within treatment arms.

Line 29

According to The World Bank, China is an upper middle-income country now. However, it is still reasonable because this intervention study was conducted in 2009/2010. the authors could also mention that the study offer experiences for other low-income countries to improve the malnutrition among children.

Line 38

This sentence looks incomplete.

Line 43

The reference numeric should be followed by authors’ name (e.g. Powell et al. [14] found that …). Same problems could be seen in the whole manuscript.

Line 48

The authors mentioned “to understand the underlying causes of these discrepancies”. I don’ think the intervention study can explain this research gap.

Line 54

This question is indeed important at first glance.

 Line 59-62

“Since… observe outcomes”, I’m not sure what do the authors mean here.

Line 78

Why is the “econometric analysis”? Please be concise about this paragraph since it repeated the conclusion section.

Line 97

This manuscript was submitted after the intervention already finished for almost 10 years. Why?

Line 100

Please add the year of this prevalence data. And the reference was published in 2011, not “a recent study”.

Line 103

Please add references for WHO standards.

Line 108

Written consents or oral consents? Please be specific.

Line 118-123

The sample selection process is not clear. How many boarding students in total from 6994 students? How many boarding students provided variables of interest, such as Hb level, DDS, etc?

Line 130

How exactly did the treatment group 1 receive the intervention? Please be more specific, e.g., where these money be used within schools, in canteens or nutrition education? Did you provide any requirements or recommendations regarding how to use money? Did schoolmasters know about the aims of this intervention programs?

Treatment group 2: what is the content and procedure to increase Hb Levels in this group?

We need to know, what the control group was eating and what food and beverages they received on a common basis in order to interpret the results of all groups accordingly.

Section 2.4

Definition and derivation of exposure and outcome variables is not clear. It is better to first mention that “our outcomes were including policy-related outcomes (i.e., Hb concentration level and the incidence of anemia), nutritional status outcomes (i.e., BMI z-scores and the incidence of underweight), and diet structure outcomes (i.e., DDS)”, just as described in results section. The authors describe the details about how they measured (e.g., unit) and defined/categorized (e.g., WHO criteria) each outcome. At last what kind of covariates did you take into consideration (e.g., personal or family characteristics, area of residence, age,sex, school-identification).

Line 158

About Dietary Diversity Score:

(1)    Why did the authors choose this score to measure the dietary structure? Are there any other references using this indicators in Chinese youth population?

(2)    How did the authors calculate the core for each food group exactly, like if a participant had some specific food, then the score=1, if not=0? If so, there are only two values in each food group for all participants, why didn’t they use dichotomous variables (%) instead of continuous variables (mean)?

Line 189

Which statistical software and program did authors use to estimate the results? What statistical significance level was chosen to interpret the results?

Line 191

As I understood from the result tables, the authors have continuous outcomes (hemoglobin concentration and BMI-for-age z-scores) and category outcomes (being anemia and underweight), and I am not sure what type of the DDS outcomes were as I mentioned above. How did the authors treat different kind of outcomes with same regression model? How could these category variables be calculated as difference-in-differences setup (Line 200)?

Line 212

Nutritional outcomes (Line 212); anemia-related outcomes (line 192); policy-related outcomes (214), please use the same words.

Line 224 and line 228

About “a 12 percentage-point reduction” and also “4.1 percentage points”, since I am not sure which statistical models the authors exactly used, it is difficult to follow the results and conclusions.

Line 234

Which ethnicity? Please be specific, also dietary related restrictions/taboos.

Line 239

Typo: structure.

Line 277

What explanation exactly?

Line 314

About Table A1, (1) in the current study only boarding students were included. It is better to provide the baseline characteristics just for boarding students. (2) please also provide the total number N in each group.

Reviewer 3 Report

This article capitalizes on policy interventions to ask about the impact of subsidies for food on the food intake and health of students in rural China.  This type of research is important, as school food and nutrition policy continues to receive increasing scrutiny as an instrument to improve health.  I have a number of suggestions that I think will strengthen the article.  Please be aware that while I have expertise in school food and nutrition policy, I am less familiar with quantitative analysis, so other reviewers may have more pertinent feedback than I regarding that aspect.  

First, a couple of points about the organization of the article.  It seems to me that Lines 78-88 are results and belong in that section of the paper.  Likewise, it seems that lines 168-173, along with Table 1, are also results and should be moved.

Second, and the most extensive comments, are about clarity of the methods and presentation of the results and analysis.  

a.  This comment may be more a reflection on me than you, but please be very clear about your sampling.  I'm assuming that within the treatment schools, all children received the extra food, that the treatment wasn't targeted to some students and not others (but perhaps not all of these students contributed to data collection).  If I'm wrong and only targeted students within schools received the extra food and had data collected from them, that's important information.  So, in order to avoid any potential confusion, please clarify exactly who the sample was in relation to all students in the schools.

b.  More specific and additional information is needed about the 24-hour recall.  It appears that you only collected one 24-hour recall from the students.  On line 168, you indicate the dietary information was "observed in the evaluation period," but you need to let us know exactly when the data were collected (e.g., what month).  Typically, when researchers report on collecting dietary data, they indicate who collected the data (e.g., were they trained nutrition professionals or . . ., if they used any aids to help students with their recall, the method they used for collecting the data (there are different strategies), and in your case, whether you collected frequencies of consumption only or frequencies and amounts (Lines 286-288 seem ambiguous - it's best to be clear in the methods section and throughout the paper).  You have an advantage because these were not free-living students so you were aware of their possible choices.  It would help to remind readers of that fact, since it strengthens the validity of their recall.

c.  Regarding the Diversity Score (Line 158), I see you reference FAO (18) in the notes, but it would be preferable to indicate this reference in the actual text, otherwise the reader might miss it.  Related to the analysis, did you consider creating an Iron-rich foods group comparable to the Vitamin A rich group used by the FAO?  The advantage of that would be that iron-rich foods cut across most food groups, except dairy, so grouping them could provide a stronger indicator than the way these data were presented.  If not, might that be something to consider in the future (e.g., suggest for future research)?  By the way, citrus fruits (or at least oranges) are a poor source of iron and should be removed as an example (line 262) - they are an excellent source of Vitamin C, which you mention on line 264.  On the other hand, tofu is a good source of iron - is it part of the cuisine of this region/diet of these students, i.e., would it be useful to add as an example? 

d.  I recommend you include additional data to help readers gain a clear picture of what happened with these students, especially as you are relating this research to policy.  And first, we need to know the number of students in each group, the control and two treatment groups.  That information should be included whenever their results are provided (e.g., table 1).  Second, for each group, it would strengthen the article if you reported (e.g., Revise Table 2, Line 233):

iron level before and after 

anemia number of students before and after

dietary information before and after (or as much as is known)

It would also help to know the actual standard being used (Line 103 - WHO standards - what are they)?  With this additional information, for example, we would know the amount of money the treatment 2 schools earned by reducing the number of students with anemia.

Third, in the discussion, it is also critically important to identify the potential drawbacks of targeted policies such as the iron-specific treatment.  For example, although it did not seem to occur here, unscrupulous schools might provide only the anemic students with extra food or might only offer iron rich food, not food such as extra dairy products.  Likewise, although you mention (Lines 294-297) that perhaps this type of policy approach could be applied to other situations, I would add a caution.  Increasing iron intake, Vitamin A intake, or protein intake for example are specific targets.  Weight, on the other hand, is much more complex.  Again, might an unintended consequence of such a policy be that children are discouraged from physical activity and given lots of high calorie foods to eat (as a worst-case scenario)?  As mentioned, these policy recommendations must be considered more fully, not just recommended carte blanche.  

It would also help to discuss why these results occurred in this research, when results from some of the other research mentioned early in the article has been more ambivalent - e.g., longer study period?  students' anemia was more severe in China????  That way, the paper is tied together from beginning to end.

Fourth, just a couple of comments about language as in general, this article is well written.  for example, Line 119 contains informality ("After all,  . . .), which is best to delete.  The way the sentence begins on line 213, "Turn first . . . " it seems like it is written in the second person when all other writing is not.  

Round 2

Reviewer 2 Report

My main concern still is that the authors are not able to describe the intervention activities/content that relate to the outcomes. This is like a black box. Also, this does not provide new knowledge for the development of public health strategies for the future, thus what is the message the authors want to transfer?

Author Response

  1. The authors are not able to describe the intervention activities/content that relate to the outcomes. This is like a black box.

Response: Thanks for pointing out the lack of clarity in our description. The activities are, in fact, reflected in students’ consumption frequencies of a set of FAO-recommended food groups (e.g. Grains, Tubers, Vegetables, etc. see Table 1). This is why we made use of students’ 24-hour dietary recall data to perform our analysis. In the muscript, we provided information on both the raw (unconditional) values of their consumption ferquencies (both baseline values and follow-up values; see Table 2) and the impacts of our treatments on the consumption frequencies of the ten food groups (see Table 4). Also, as mentioned in the manuscript (lines 125-126), to capture students’ consumption activities more accurately, we focus on boarding students, whose entire meal plan was controlled by the school. Put differently, the use of the 24-hour dietary recall data (as reflected in Tables 2 and 4) is our strategy to crack the “black box” open.

More specifically, we use the following conceptural framework, which involves three sets of ourcomes, to help with openning the black box, as summarized in Figure 1. The first set of outcomes (anemia-related) are natural for capturing the effectiveness of the intervention – if the specific policy target of “anemia reduction” (and the incentives attached) did work, one would expect to see at least some reductions in the prevalence of anemia among students in treatment group 2 (i.e. to capture “intended outcomes”). The second set of outcomes (overall nutritional status) help to assess whether these changes translated into improvements in their overall nutritional status (i.e. to capture spillovers or “unitended outcomes”). Most importantly, the third set, that on dietary structure (reflected in students’ 24-hour recall consumption activities), is used to capture the possible working channels (i.e. to open the “blackbox”).

Figure 1. Conceptual Framework.

In revision, we put more emphasis on the points discussed above to avoid further confusion.

  1. Also, this does not provide new knowledge for the development of public health strategies for the future, thus what is the message the authors want to transfer?

Response: Thanks for pointing out the lack of clarity in our conclusion. By discovering that policy targets with different levels of specificity (with the incentives attached to them) can lead to different dietray structures (as reported in Tables 2 and 4) and thus nutrional outcomes (as reported in Tables 3 and A2), our analysis does provide new knowledge for the development of public health strategies for the future. Note first that the estimation of treatment effects on the DDS (diet diversity index) itself provides some new information, as the DDS was not widely used until very recently. Second, the suggestion of “make the policy target more specific (e.g. as we discovered, changing “malnutrition reduction” to “anemia reduction”) and design proper incentives” is based on a new discovery, and it has everything to do with public health strategies. Third, we also discovered positive spillovers from the treatment with a specific policy target (“anemia reduction”) – not only did this treatment significantly reduce anemia prevalence (which is the intended outcome), but it also reduce the prevalance of underweight (which is an unitended yet desriable outcome). Finally, we provide new information by way of identifying room for furture research, such as to investigate the impacts of other specific targets (e.g. tied to “weight”, “protein intake” and “Vitamin A intake”, and other nutrition and health indices) and other incentive schemes (e.g. different amouts of bonus money). We envision some of these police targets will also generate positive spillovers as treatment 2 in our intervention, while some will lead to negative spillovers, but their real effects are left for future research to confirm.

In revision, we put more emphasis on the points discussed above to avoid further confusion.